# Density Functional Theory for Buckyballs within Symmetrized Icosahedral Basis

**DOI:** 10.3390/nano13131912

**Published:** 2023-06-23

**Authors:** Chung-Yuan Ren, Raj Kumar Paudel, Yia-Chung Chang

**Affiliations:** 1Department of Physics, National Kaohsiung Normal University, Kaohsiung 824, Taiwan; cyren@nknu.edu.tw; 2Research Center for Applied Sciences, Academia Sinica, Taipei 115, Taiwan; rajupdl6@gate.sinica.edu.tw; 3Molecular Science and Technology, Taiwan International Graduate Program, Academia Sinica, Taipei 115, Taiwan; 4Department of Physics, National Central University, Chungli 320, Taiwan; 5Department of Physics, National Cheng-Kung University, Tainan 701, Taiwan

**Keywords:** density functional theory (DFT), C_60_, buckyball, symmetrized icosahedral basis, optical absorption spectrum

## Abstract

We have developed a highly efficient computation method based on density functional theory (DFT) within a set of fully symmetrized basis functions for the C_60_ buckyball, which possesses the icosahedral ($I_{h}$) point-group symmetry with 120 symmetry operations. We demonstrate that our approach is much more efficient than the conventional approach based on three-dimensional plane waves. When applied to the calculation of optical transitions, our method is more than one order of magnitude faster than the existing DFT package with a conventional plane-wave basis. This makes it very convenient for modeling optical and transport properties of quantum devices related to buckyball crystals. The method introduced here can be easily extended to other fullerene-like materials.

## 1. Introduction

Since the discovery of “buckminsterfullerene” (also known as the buckyball) in 1985 [1], the highly symmetric C_60_ buckyball has attracted tremendous attention in the scientific community. C_60_ possesses icosahedral ($I_{h}$) point-group symmetry [2] with 120 symmetry operations. Buckyballs and other fullerenes of similar properties [3,4] have many potential applications that include quantum computing [5,6,7,8], biosensing [9], lubricants [10], ultrahigh strength materials [11], and nanoscale optoelectronic devices [12]. 

Buckyballs can also form single crystals and they have decent mobility for device applications [13,14]. Many fullerenes can display superconductivity at relatively high temperatures. It was observed that Cs-doped C_60_ single crystals have a superconducting transition temperature at $T_{C}=$ 40 K [15] and several alkali metal-doped C_60_ compounds exhibit $T_{C}$ in the range between 19 K and 47 K [16]. The existence of high-Tc superconductivity in fullerenes is likely caused by the strong electron–phonon interaction, but detailed microscopic theory for understanding this is still not available. The theoretical development for such an important problem is mainly hindered by the complexity of the system, which requires heavy computation and highly sophisticated theoretical analyses. Thus, an effort to significantly reduce the computation effort for calculating the electronic states in such systems is warranted.

For the C_60_ buckyball, which is a truncated icosahedron [3,17,18], it is natural to choose a basis set consisting of products of spherical harmonic functions [$Y_{lm}(\Omega )]$ and localized basis functions along the radial direction. The point group theory can be used to find proper linear combinations of spherical harmonics to form symmetry-adapted basis functions (SABFs) which transform according to irreducible representations (IRs) of the icosahedral group $I_{h}$ [18]. It has been pointed out that nanoscale systems with high point-point symmetry can be more efficiently solved by using symmetrized basis functions [19]. The symmetry-adapted basis functions have also been used in some quantum chemistry packages [20] for calculating the electronic properties of molecules. Thus, we can take advantage of the 120 symmetry operations of the $I_{h}$ group to construct symmetrized basis functions that are convenient for modeling related solid-state systems. The saving in computation time will be significant since the dimension (*N*) of the Hamiltonian matrix for each symmetry type will be orders of magnitude less than the full Hamiltonian matrix, and the computation time needed for diagonalization scales like *N*^3^ if a full-matrix solver is used. Furthermore, studying the excitation properties involves an exciton, which consists of an electron and a hole. To calculate the excitation spectra, one needs to solve the Bethe–Salpeter equation [21]. The number of possible product states for solving the Bethe–Salpeter equation scales like *N*^2^. Thus, using a fully symmetrized basis would speed up the computation of excitation properties by another order of magnitude or more. Therefore, developing a density functional theory (DFT) computation package specially designed for high-symmetry systems such as crystals made of buckyballs and other fullerene-like materials [22,23] will be worthwhile. 

This paper aims to demonstrate the usefulness of symmetrized basis in the development of DFT for high-symmetry systems such as fullerenes and related crystals for application in solid-state devices. The standard analytic procedure for finding suitable linear combinations of spherical harmonic functions based on the group theory can be tedious. In this paper, we introduce a simple scheme to utilize a computation method to extract the coefficients in the symmetrized basis which transform according to the IRs of the underlying point group. It is convenient to use, and it can avoid errors introduced by using different conventions for defining the basis functions. We use the C_60_ buckyball as an example to illustrate the advantages of the method and discuss how to use these convenient basis functions to study related fullerene-like materials.

## 2. Materials and Methods

### 2.1. DFT Based on Symmetrized Angular Functions Augmented by Radial B-Splines

The C_60_ molecule has two types of C–C bonds. For simplicity, all carbon atoms are located at the ideal positions of the buckyball, with the equivalent bond length of 1.4 Å [24]. A schematic diagram of the C_60_ molecule is depicted in Figure 1 [25].

In density functional theory (DFT), the Kohn–Sham Hamiltonian for an electron in the C_60_ molecule is written as:(1)$$H=-{\nabla^{2}+V}_{loc}+{\hat{V}}_{nl}$$
where we have adopted the atomic units throughout the paper with energy measured in Rydberg (Ry) and distance in bohr. In the right-hand side of Equation (1), the first, second, and third term describes the kinetic energy, the local pseudopotential, and the nonlocal pseudopotential, respectively. The local pseudopotential consists of three terms:(2)$$\begin{array}{r} V_{loc}\left(r\right)=V_{ion}\left(r\right) \end{array}+V_{H}\left(r\right)+V_{xc}\left(r\right)$$
where the first term describes the ionic local potential with
(3)$$V_{ion}\left(r\right)=\sum_{\sigma }V_{I}^{\;}\left(r-\tau_{\sigma }\right)$$
in which $\tau_{\sigma }$ denotes the position of different C atoms in the buckyball. For simplicity, we adopt the norm-conserving pseudopotential (NCPP) developed by Goedecker, Teter, and Hutter (GTH) [26]. The ultrasoft pseudopotential (USPP) developed by Vanderbilt [27] can also be adopted in our current approach. However, the implementation of the projector augmented part used to reduce the number of plane waves (or spherical harmonics here) in the basis to achieve faster convergence in the calculation will require more effort. Since we aim to demonstrate the usefulness of the symmetrized icosahedral basis to facilitate the DFT calculation of high-symmetry systems, such as C_60_ and related crystals, we choose to start with a simpler scheme.

In the GTH approach, $V_{I}^{\sigma }\left(r\right)$ can be well described by a simple analytic form [26]
(4)$$V_{I}^{\;}\left(r\right)=-\frac{Z_{ion}}{r}erf(\sqrt{\alpha_{0}}r)+\sum_{j=0}^{3}D_{j}r^{2j}e^{-\alpha_{0}r^{2}}$$
where erf denotes the error function and $Z_{ion}$ is the ionic charge. $\alpha_{0}$ and $D_{j}$ are fitting parameters for the C atom. The Fourier transform of $V_{I}^{\;}\left(r\right)$ is also given by a simple form [26]
(5)$${\tilde{V}}_{I}^{\;}\left(k\right)=-\frac{4\pi Z_{ion}}{V}\frac{e^{-k^{2}/4\alpha_{0}}}{k^{2}}+\frac{1}{V}\sum_{j=0}^{3}D_{j}{\left(\frac{-\partial }{\partial \alpha_{0}}\right)}^{j}{\left(\frac{\pi }{\alpha_{0}}\right)}^{3/2}e^{-k^{2}/4\alpha_{0}}$$
where V is the sample volume. $V_{H}\left(r\right)$ in Equation (2) denotes the self-consistent Hartree potential and the last term in Equation (1) denotes the exchange-correlation potential, which is deduced from the Monte Carlo results calculated by Ceperley and Alder [28] and parametrized by Perdew and Zunger [29]. The DFT effective potential is determined self-consistently until its root-mean-square change is less than 10^−6^ Ry.

The nonlocal pseudopotential (${\hat{V}}_{nl})$ in the GTH approach is given by
(6)$${\hat{V}}_{nl}=\sum_{slm}h_{s}^{l}\left|\left.\beta_{lm}^{s}\right\rangle \left\langle \beta_{lm}^{s}\right.\right|$$
where
(7)$$\beta_{lm}^{s}\left(r\right)=p_{s}^{l}\left(r\right){Y_{lm}(\Omega )=p_{s}^{l}\left(r\right)P}_{lm}\left(\theta \right)e^{im\varphi }$$
for l=0.1 and $\left|m\right|\leq l.$ s=1.2 denotes different *β* functions used for each angular momentum, l. The normalized $p_{s}^{l}\left(r\right)$ functions have simple analytic forms as given in [26]. $h_{s}^{l}$ are energy parameters. The projection of $\beta_{lm}^{s}$ function in the wave–vector space reads
(8)$$\left\langle k|\beta_{lm}^{s}\right\rangle =\frac{4\pi }{\sqrt{V}}{\left(-i\right)}^{l}\int_{0}^{R_{C}}drd{\Omega r}^{2}p_{s}^{l}\left(r\right){Y_{lm}(\Omega )e}^{-ik\cdot r}=\frac{1}{\sqrt{V}}{\tilde{p}}_{l}^{s}{(k)Y}_{lm}\left(\hat{k}\right)$$
(9)$$\mathrm{where}\;{\tilde{p}}_{l}^{s}\left(k\right)=4\pi {\left(-i\right)}^{l}\int_{0}^{\infty }drr^{2}p_{s}^{l}\left(r\right)j_{l}\left(kr\right)$$
which can also be expressed in a simple analytical form [26].

### 2.2. Generation of Symmetrized Angular Basis Functions

To take advantage of the full point-group symmetry, we first construct the C_60_-adapted symmetrized angular basis functions via a suitable linear combination of the spherical harmonics $Y_{lm}^{\;}\left({\hat{r}}_{}\right).$ We define the symmetrized angular functions (called icosahedral harmonics) with symmetry type (Γ,v) compatible with angular momentum l as
(10)$${K_{l}^{\Gamma v}\left(\Omega \right)=\sum_{m}C_{lm}^{\Gamma v}Y}_{lm}\left(\Omega \right)$$
(11)$$\;\mathrm{where}\;C_{lm}^{\Gamma v}=\frac{n(\Gamma )}{h}{\sum_{\Lambda }\Gamma_{vv}\left(\Lambda \right)D}_{mm}^{\left(l\right)}\left(\Lambda \right)$$
(12)$$\;\mathrm{and}\;D_{mm^{\prime }}^{\left(l\right)}\left(\Lambda \right)=\int d\Omega Y_{lm}^{*}\left(\Omega \right)Y_{lm^{\prime }}\left(\Lambda^{-1}\Omega \right).$$

Here, *Λ* denotes the 120 symmetry operations in the $I_{h}$ group and n(Γ) is the dimension of the irreducible representation Γ and h is the order of the point group. The index *v* labels the degenerate partners of an IR. One straightforward way to find the symmetrization coefficients $C_{lm}^{\Gamma v}$ is to use the projection method, commonly described in textbooks [30]. However, it requires the knowledge of transformation matrices $\Gamma_{\;}^{i}$(Λ) of the spherical harmonics for each symmetry operator *Λ* associated with the *i*-th IR of the point group. For $I_{h}$ group, the above procedure can be quite tedious. Although these coefficients for the $I_{h}$ group can be obtained from the projection method [31], it still requires a lot of effort to implement these coefficients in the current DFT code.

Here, we adopt a more practical approach. We obtain these symmetrization coefficients via diagonalization of an effective Hamiltonian matrix within a minimum basis set of the form {${e^{-\alpha r}Y}_{lm}\left(\Omega \right)\}$ for each fixed index of angular momentum, *l*. The effective potential is taken to be of the form:(13)$$V_{eff}\left(r\right)=\sum_{\sigma }V_{I}(r-\tau_{\sigma })$$
where $\tau_{\sigma }$ denotes the positions of 60 carbon atoms in C_60_, $V_{I}(r$) can be any model potential. Here, we simply choose $V_{I}(r$) to be the GTH atomic local pseudopotential adopted in the current approach. The matrix element of $V_{eff}\left(r\right)$ in the subspace {$Y_{lm}\left(\Omega \right)\}$ with a fixed *l* is given by
(14)$$\left\langle lm|V_{eff}|lm\prime \right\rangle =\sum_{\sigma }\int Y_{lm}^{*}\left(\hat{r}\right)V_{I}(r-\tau_{\sigma }{)Y}_{lm^{\prime }}\left(\hat{r}\right).$$

Obviously, $V_{eff}$ obeys the same point group symmetry as C_60_. Thus, the eigenvectors of $V_{eff}$ defined in the subspace {$Y_{lm}\left(\Omega \right)\}$ with a fixed *l* will transform according to the IRs of the $I_{h}$ group. From the degeneracy (g) of the corresponding eigenvalues, we can identify the possible irreducible representations. For example, if g=1,4,5, the states must belong to the $\Gamma_{1}\left(A\right),\Gamma_{4}\left(G\right),\;and\;\Gamma_{5}\left(H\right)$ representations, respectively. If g=3, the states must belong to either the $\Gamma_{2}\left(F_{1}\right)\;or\;\Gamma_{3}\left(F_{2}\right)$ representation. To pin down the precise symmetry type of each eigenstate, we simply evaluate the coupling matrix element between the state and a known basis state belonging to the possible representations as given in [32,33]. When the matrix element is nonzero, the symmetry type is identified. The above method can be easily applied to systems with any other point group. The symmetrized basis functions with lowest *l* for the five irreducible representations of the $\;I_{h}$ group with either even or odd parity are listed in Table 1.

### 2.3. The Choice of Basis Functions

To simplify the computation effort, we choose the B-spline [34] augmented icosahedral harmonics (BAIHs) as basis functions for the DFT calculation of C_60_ buckyball. The BAIHs are defined as
(15)$$\Phi_{nl\Gamma v}\left(r\right)=B_{n}(r)K_{l}^{\Gamma v}\left(\Omega \right)=\frac{1}{\sqrt{V}}\sum_{k}\sum_{m}C_{lm}^{\Gamma v}F_{nlm}\left(k\right)e^{ik\cdot r}$$
with
(16)$$F_{nlm}\left(k\right)=\frac{1}{\sqrt{V}}\int d^{3}re^{-ik\cdot r}B_{n}(r)Y_{lm}\left(\hat{r}\right)$$
where $B_{n}(r)$ denotes the B-spline functions of a suitable order [29].
(17)$$\;\mathrm{Using}\;e^{-ik\cdot r}=4\pi \sum_{lm}i^{-l}j_{l}\left(kr\right)Y_{lm}(\hat{k})Y_{lm}^{*}(\hat{r})$$
(18)$$\;\mathrm{we}\;\mathrm{have}\;F_{nlm}\left(k\right)=\frac{4\pi }{\sqrt{V}}i^{-l}Y_{lm}(\hat{k})I_{nl}(k),$$
(19)$$\;\mathrm{where}\;I_{nl}\left(k\right)=\int drj_{l}(kr)r^{2}B_{n}(r)$$

For each fixed B-spline basis $B_{n}\left(r\right),$ we can take linear combinations of $Y_{lm}(\Omega )$ to construct symmetrized basis functions that transform according to the irreducible representations of an icosahedral group, *I* (or $I_{h}=i⊗I$, where i denotes inversion).

To make sure that the degenerate partners in each irreducible representation transform in the same way as in the corresponding basis functions for lower *l*, we do the following. Let $K_{l_{1}}^{\Gamma v}\left(\Omega \right)$ denote the symmetrized basis function for the *v*-th degenerate partner for the Γ representation, and $l_{1}$ denotes the lowest possible *l* for this representation. We can take linear combinations of states $\left.\;|l;\Gamma v^{\prime }\right\rangle$ such that
$$\sum_{v^{\prime }}\left\langle l_{1},\Gamma v|H_{eff}|l,\Gamma v^{\prime }\;\right\rangle Z_{v^{\prime }v^{\prime \prime }\;}^{\Gamma }=\delta_{vv^{\prime \prime }}$$

Thus, *Z* matrix is proportional to the inverse of the $n\left(\Gamma \right)$-dimensional matrix with ($v,v^{\prime })$ elements given by $\left\langle l,\Gamma v|H_{eff}|l_{1},\Gamma v^{\prime }\right\rangle .$ Using this simple procedure, we can obtain coefficients for symmetrized basis states for all higher *l* needed. With this approach, we obtained the symmetry coefficients $C_{lm}^{\Gamma v}$ for the 5 irreducible representations ${\Gamma_{1}\left(A\right),\Gamma }_{2}\left(F_{1}\right),\Gamma_{3}\left(F_{2}\right){,\Gamma }_{4}\left(G\right),and\;\Gamma_{5}\left(H\right)$ of $I_{h}$ group for *l* up to 55.

### 2.4. The Matrix Elements of Local Pseudopotential

Since $V_{ion}\left(r\right)$ should be invariant under all symmetry operations of the $I_{h}$ group, we can write $V_{ion}\left(r\right)$ in the form $V_{ion}\left(r\right)=\sum_{l}{\tilde{V}}_{L}^{ion}{(r)K}_{l}^{\Gamma_{1g}}\left(\Omega \right)$ with
(20)$${\tilde{V}}_{l}^{ion}\left(r\right)=\sum_{k}V_{c}\left(k\right)\int d\Omega \;{e^{-ik\cdot (r-\tau_{\sigma })}K}_{l}^{\Gamma_{1g}}\left(\Omega \right)+\sum_{j=0}^{3}D_{j}r^{2j}e^{-\alpha_{0}\left(r^{2}+\tau_{\sigma }^{2}\right)}\sum_{\sigma ,m}\int d\Omega \;e^{{2\alpha }_{0}{r\cdot \tau }_{\sigma }}K_{l}^{\Gamma_{1g}}\left(\Omega \right)\;=4\pi i^{-l}\sum_{\sigma ,m}C_{lm}^{\Gamma_{1g}}\left[\int \frac{dk}{2\pi^{2}}e^{-k^{2}/4\alpha_{0}}{j_{0}\left(k\tau_{\sigma }^{\;}\right)j}_{l}\left(kr\right)+\sum_{j=0}^{3}{D_{j}r^{2j}e}^{-\alpha_{0}\left(r^{2}+\tau_{\sigma }^{2}\right)}j_{l}\left({2i\alpha }_{0}\tau_{\sigma }^{\;}r\right)\right]Y_{lm}^{*}\left({\hat{\tau }}_{\sigma }\right)$$
where $V_{c}\left(k\right)=-\frac{4\pi Z_{ion}}{V}\frac{e^{-k^{2}/4\alpha_{0}}}{k^{2}}.$

For the ideal buckyball, the magnitude of $\tau_{\sigma }^{\;}$ is the same for all 60 carbon atoms. Note that there is an overflow problem in $j_{l}\left({2i\alpha }_{j}\tau_{\sigma }^{\;}r\right)$ when $\alpha_{j}$ is large, since
$$j_{0}\left({2i\alpha }_{j}\tau_{\sigma }^{\;}r\right)=\frac{sin\left({2i\alpha }_{j}\tau_{\sigma }^{\;}r\right)}{{2i\alpha }_{j}\tau_{\sigma }^{\;}r}=\frac{e^{{2\alpha }_{j}\tau_{\sigma }^{\;}r}-e^{{-2\alpha }_{j}\tau_{\sigma }^{\;}r}}{{4\alpha }_{j}\tau_{\sigma }^{\;}r}.\;\mathrm{However},$$
$$e^{-\alpha_{j}(r^{2}+\tau_{\sigma }^{2})}j_{0}\left({2i\alpha }_{j}\tau_{\sigma }^{\;}r\right)=\frac{e^{{-\alpha }_{j}{\left(r-\tau_{\sigma }^{\;}\right)}^{2}}-e^{{-\alpha }_{j}{\left(r+\tau_{\sigma }^{\;}\right)}^{2}}}{{4\alpha }_{j}\tau_{\sigma }^{\;}r}\;\mathrm{is}\;\mathrm{well}\;\mathrm{behaved}.\mathrm{Similarly}\mathrm{we}\;\mathrm{have}$$
$$\;e^{-\alpha_{j}(r^{2}+\tau_{\sigma }^{2})}j_{1}\left({2i\alpha }_{j}\tau_{\sigma }^{\;}r\right)=\;\frac{e^{{-\alpha }_{j}{\left(r-\tau_{\sigma }^{\;}\right)}^{2}}-e^{{-\alpha }_{j}{\left(r+\tau_{\sigma }^{\;}\right)}^{2}}}{8i{{(\alpha }_{j}\tau_{\sigma }^{\;}r)}^{2}}-\frac{e^{{-\alpha }_{j}{\left(r-\tau_{\sigma }^{\;}\right)}^{2}}+e^{{-\alpha }_{j}{\left(r+\tau_{\sigma }^{\;}\right)}^{2}}}{{4\alpha }_{j}\tau_{\sigma }^{\;}r}$$
where we have used $j_{1}\left(z\right)=sin(z)/z/z-cos(z)/z$.

Using the recursion relation for spherical Bessel functions, we have
$${\tilde{j}}_{n+1}\left(z\right)=(2n+1){\tilde{j}}_{n}(z)/z-{\tilde{j}}_{n-1}(z),$$
where ${\tilde{j}}_{n}(z)=f(z)j_{n}(z)$ with $f(z)=e^{-\alpha_{j}(r^{2}+\tau_{\sigma }^{2})}$ independent of *n*.

$V_{H}\left(r\right)=\int d^{3}r^{\prime }\rho (r^{\prime })/\left|r-r^{\prime }\right|$ denotes the Hartree potential, where $\rho \left(r\right)$ denotes the charge density. Since the wavefunctions are expanded in terms of BAIHs, we can write
(21)$$\rho \left(r\right)=\sum_{j}{\left|\Psi \left(r\right)\right|}^{2}=\sum_{L}{\tilde{\rho }}_{L}\left(r\right)K_{L}^{\Gamma_{1g}}\left(\Omega \right)=\sum_{LM}{\tilde{\rho }}_{L}\left(r\right)C_{LM}^{\Gamma_{1g}}Y_{LM}^{\;}\left(\Omega \right)$$

Here, the charge density transforms according to the $\Gamma_{1g}$ representation. Thus, we have ${\tilde{\rho }}_{L}\left(r\right)=\int d\Omega K_{L}^{\Gamma_{1g}}\left(\Omega \right)\rho \left(r\right)$.

Due to the high symmetry, the integral only has to cover 1/120 of the whole solid angle as shown by half the surface area enclosed by green lines in Figure 1. Using the expansion
(22)$$\frac{1}{\left|r-r^{\prime }\right|}=\sum_{lm}\frac{4\pi }{2l+1}\frac{r_{<}^{l}}{r_{>}^{l+1}}Y_{lm}^{\;}({\hat{r}}_{})Y_{lm}^{*}({\hat{r}\prime }_{})$$
we obtain
(23)$$V_{H}\left(r\right)=\sum_{lm}\left[\frac{4\pi }{2l+1}\int r^{\prime 2}dr\prime {\tilde{\rho }}_{l}(r\prime )\frac{r_{<}^{l}}{r_{>}^{l+1}}\right]C_{lm}^{\Gamma_{1g}}Y_{lm}^{\;}({\hat{r}}_{})\equiv \sum_{l}{\tilde{V}}_{l}^{H}\left(r\right)K_{l}^{\Gamma_{1g}}\left(\Omega \right)$$
where $r_{<}=\min\left(r,r\prime \right)and\;r_{>}=\max\left(r,r\prime \right),$ and
(24)$${\tilde{V}}_{l}^{H}\left(r\right)=\left[\frac{4\pi }{2l+1}\int r^{\prime 2}dr\prime {\tilde{\rho }}_{l}(r\prime )\frac{r_{<}^{l}}{r_{>}^{l+1}}\right]$$

Here, $V_{xc}\left(r\right)$ denotes the exchange-correlation potential. In each iteration, we shall expand $V_{xc}\left(r\right)$ in terms of spherical harmonics in the form
$$V_{xc}\left(r\right)=\sum_{L}{\tilde{V}}_{L}^{xc}\left(r\right)K_{L}^{\Gamma_{1}}\left(\Omega \right)=\sum_{LM}{\tilde{V}}_{L}^{xc}\left(r\right)C_{LM}^{\Gamma_{1g}}Y_{LM}\left(\Omega \right)$$

So, ${\tilde{V}}_{L}^{xc}\left(r\right)=\int d\Omega K_{L}^{\Gamma_{1g}}\left(\Omega \right)V_{xc}\left(r\right),$ where the integral can be done efficiently by using the symmetry property. 

Then, the matrix elements read
(25)$$\left\langle \Phi_{nl\Gamma v}|V_{loc}|\Phi_{n^{\prime }l^{\prime }\Gamma v}\right\rangle =\sum_{LM}V_{L}^{nn\prime }\left\langle lm|LM;l\prime m\prime \right\rangle C_{LM}^{\Gamma_{1g}}=\sum_{L}V_{L}^{nn\prime }S_{L}$$
(26)$$\;\mathrm{where}\;V_{L}^{nn^{\prime }}=\int \left[{{\tilde{V}}_{L}^{ion}\left(r\right)+\tilde{V}}_{L}^{H}\left(r\right)+{\tilde{V}}_{L}^{xc}\left(r\right)\right]r^{L+2}\;B_{n}(r)B_{n\prime }(r)dr,$$
and $S_{L}=\sum_{M,m,m^{\prime }}C_{lm}^{\Gamma v*}\left\langle lm|LM;l\prime m\prime \right\rangle C_{l^{\prime }m^{\prime }}^{\Gamma v}{.\;C}_{LM}^{\Gamma_{1g}}$ can be evaluated for all *L* (up to 55) before the DFT calculation.

### 2.5. Nonlocal Pseudopotential

The matrix elements of ${\hat{V}}_{nl}$ within the BAIH basis can be evaluated efficiently due to its separable form. We have
(27)$$\left\langle \Phi_{nl\Gamma v}|{\hat{V}}_{nl}|\Phi_{n^{\prime }l^{\prime }\Gamma v}\right\rangle =\sum_{slm}h_{s}^{l}\left\langle \Phi_{nl\Gamma v}|\beta_{lm}^{s}\right\rangle \left\langle \beta_{lm}^{s}|\Phi_{n^{\prime }l^{\prime }\Gamma v}\right\rangle$$
with
(28)$$\left\langle \Phi_{nl\Gamma v}|\beta_{lm}^{s}\right\rangle =\sum_{k}\left\langle \Phi_{nl\Gamma v}|k\right\rangle \left\langle k|\beta_{lm}^{s}\right\rangle =\frac{1}{2\pi^{2}}i^{-l}C_{lm}^{\Gamma v}\int k^{2}dkI_{nl}(k){\tilde{p}}_{l}^{s}(k)$$
where $I_{nl}(k)$ is given in Equation (19) and ${\tilde{p}}_{l}^{s}\left(k\right)$ in Equation (9).

A total of 18 B-splines, defined over a range of 12 a.u., are used to expand the radial wavefunction. The maximum angular moment *L* of the symmetry-adapted basis function (SABF) used is 45. The numbers of SABFs for these 10 IR’s are listed in Table 2. Due to the small dimension of the Hamiltonian matrix for each symmetry type considered, the diagonalization of the Hamiltonian can be done efficiently with a direct solver.

## 3. Results and Discussions

### 3.1. Energy Levels

We found that there are 32 distinct energy levels (not including the degeneracy factor) for the occupied levels (labeled by $E_{v,j})$ for the C_60_ molecule. Table 3 displays the number of occurrences (*n*) for each symmetry type or the occupied energy levels. We also list the corresponding values of n×g in the last row for each symmetry type to account for the level degeneracy (g). Note that the sum of n×g for all occupied levels is equal to 120 since each C atom contributes two electrons to the occupied levels (or valence states). 

For comparison purposes, we also performed calculations of the C_60_ buckyball by using the Quantum ESPRESSO (QE) plane-wave-based package [35] with the NCPP option; the exchange-correlation functional parametrized by Perdew and Zunger [29] (same as the one used in the current approach) was used. In the QE calculation, a cubic supercell with 20 Å along each side is chosen for the calculation. Thus, we are modeling an artificial buckyball crystal with the QE approach, instead of a single molecule as considered in our current code. We have checked the suitability of the vacuum length used in the QE calculation and found that the results concerned here are not significantly altered when the cell size is varied between 15 Å and 20 Å. Since C_60_ is not electrically polarized, based on previous calculations on graphene nanoribbons [36], a vacuum space of ~10 Å is enough. So, a cell length of 16–20 Å for C_60_ is typical (10–14 Å of vacuum + 6 Å for the buckyball). The energy cutoff of 70 Ry was used (typical for the NCPP adopted). 

We show the comparison of results obtained by the current method and those by using both the QE package [35] and Gaussian 16 package [20] for the inter-level energy spacings of the highest 20 occupied (valence) levels and lowest 3 unoccupied (conduction) levels in Table 4 The HOMO (highest occupied molecular orbital) level obtained by the current calculation is at −2.71 eV. We define the energy-level differences ${∆c}_{j}=E_{c,j+1}-E_{c,j}$ and ${∆v}_{j}=E_{v,j+1}-E_{v,j}$, for the inter-level energy spacings between two consecutive unoccupied (conduction) levels and occupied (valence) levels, respectively, while the band gap energy is given by ${E_{g}=E}_{c,1}-E_{v,1}$. Here, j=1 denotes the topmost valence (lowest conduction) energy level, and higher j indicates energetically decreasing (increasing) levels for valence (conduction) levels. The corresponding symmetry types (IRs) are also indicated in parentheses. 

As can be seen from Table 4, the results obtained for the bound states in the molecule (which include all occupied levels and some low-lying conduction levels) by the present method agree quite well with those results obtained by QE and reasonably well with Gaussian 16. In the calculation with Gaussian orbitals, we selected the VWN option (also within the local density approximation) [37,38]. We note that the Gaussian package uses the all-electron approach instead of the pseudopotential method. Thus, there is more deviation between our results and Gaussian 16 results. For the high-lying conduction states (which correspond to unbound states of the molecule), we see some deviation of our results from those obtained by QE and Gaussian 16.

This discrepancy is due to the different boundary conditions imposed in different approaches. In our approach and Gaussian 16, we consider only an isolated C_60_ molecule, while an artificial C_60_ solid was considered in the supercell approach used by QE. For the unbound conduction states, there will be a strong overlap between states derived from neighboring C_60_ molecules in the artificial solid. Thus, these states will have significant dispersion. Namely, these energy levels are **k**-dependent, where **k** is the wavevector of the C_60_ solid. For the current approach, we choose a finite range for the B-spline basis functions. This effectively introduces a quantum barrier for these unbound states. The discrete levels we obtained for these unbound states represent a discretized sampling of the continuum states. Thus, the energy spacing of these unbound states will depend on the range of the knot sequence chosen for the B-spline basis functions. However, this finite-size sampling of continuum states can still give a reasonable description of the optical excitation spectrum even into the continuum region, as long as the energy spacings are smaller than the line broadening used to mimic the absorption spectra. That means the sampling is dense enough to capture the main features of the optical excitation spectrum. For Gaussian 16, there is no rigid boundary. However, the energy spacings between unbound states depend sensitively on the number of Gaussian orbitals chosen and the values of exponents used.

To do a bench-mark comparison in computation speed we run both the current code and the QE package with a single processor. The CPU time needed to calculate all eigenstates for the C_60_ molecule is ~300 s with the current code, while it would take ~1000 s to get only the 120 occupied levels by QE. We note that the diagonalization procedure in QE was done via the conjugate gradient (CG) method. To study the optical properties of C_60_, we need to include many conduction states. If we also calculate 300 unoccupied levels by QE, the CPU time needed will increase to ~20 h (on a single processor). Therefore, our method is more than two orders of magnitude faster than the well-optimized QE package for such an application. With further optimization, the current code can be made even more efficient.

### 3.2. Optical Absorption Spectrum

In this section, we calculate the optical absorption spectrum of the C_60_ buckyball, while neglecting the excitonic effect and compare it with the corresponding results obtained by QE. The optical absorption spectrum is proportional to the imaginary part of the dielectric response function as given by [39]
(29)$$\varepsilon_{2}\left(\hbar \omega \right)\propto \frac{1}{\omega^{2}}\sum_{i,j}|\left\langle v,i|\hat{e}\cdot p|c,j\right\rangle {|}^{2}\delta \left(\hbar \omega +E_{v,i}-E_{c,j}\right).$$

Here $\hat{e}$ denotes the polarization vector of the photon, $\left.|v,i\right\rangle$ and $\left.|c,j\right\rangle$ denote the *i*-th valence (occupied) state with energy $E_{v,i}$ and *j*-th conduction (unoccupied) state with energy $E_{c,j}$, respectively. p denotes the momentum operator, and ħω is the photon energy. Since the symmetry types of the eigenstates obtained by the current code are already known, we can apply the selection rules and significantly reduce the computation effort for calculating the dielectric response function, $\varepsilon_{2}\left(\hbar \omega \right)$. In this work, all eigenstates of the C_60_ buckyball are localized functions, and it is convenient to use the commutator relation $p=i\frac{m}{\hbar }[H,r]$ and obtain
(30)$$\varepsilon_{2}\left(\hbar \omega \right)\propto \sum_{i,j}{\left|\left\langle v,i|\hat{e}\cdot r|c,j\right\rangle \right|}^{2}\delta \left(\hbar \omega +E_{v,i}-E_{c,j}\right).$$

Let $\varphi_{i}^{\Gamma_{h}v_{h}}\left(r_{h}\right)$ and $\varphi_{j}^{\Gamma_{e}v_{e}}\left(r_{e}\right)$ denote the molecular orbitals (MOs) in a C_60_ buckyball with symmetry type $\Gamma_{h}v_{h}$ and $\Gamma_{e}v_{e}$ for occupied and unoccupied levels, respectively. The photon transforms like l=1 spherical harmonics which has $\Gamma_{2u}$ symmetry under the $I_{h}$ group. The selection rule imposed by applying the group theory indicates that the transition $\Gamma_{h}v_{h}\to \Gamma_{e}v_{e}$ is forbidden when the vector coupling coefficient $C\left(\Gamma_{h}v_{h},\Gamma_{2u}v,;\Gamma_{e}v_{e}\right)=0.$ Here, the vector coupling coefficient plays the same role as the Clebesh–Gordan coefficient for the products of two spherical harmonics. Namely, for a given valence state with symmetry type $\Gamma_{h}$, the final conduction state must belong to an IR, compatible with the symmetry of the product $\Gamma_{h}⊗\Gamma_{2u}$. The selection rules can be worked out by using the group theory, and they are listed in Table 5. 

Based on the Wigner–Ekart theorem [40], the dipole matrix elements $\left\langle v,i|r|c,j\right\rangle$ can be written as
(31)$$\left\langle v,i|r|c,j\right\rangle =\left\langle v,i|\left|r\right||c,j\right\rangle C\left(\Gamma_{h}v_{h},\Gamma_{2u}v,;\Gamma_{e}v_{e}\right)$$
where $\left\langle v,i|\left|r\right||c,j\right\rangle$ is called the reduced matrix element, which is independent of the indices of the degenerate partners in the initial state ($v_{h})$ and final state ($v_{e})$, as well as the polarization of the photon (indexed by *v*). Thus, for each allowed transition between two manifolds, we only have to evaluate the reduced matrix element once and immediately obtain all related matrix elements with saving in computation time of $n_{v}\times n_{c}\times 3$ fold, where $n_{v}$ and $n_{c}$ are the dimensions of the $\Gamma_{h}$ and $\Gamma_{e}$ IRs, respectively. We have worked out the vector-coupling coefficients for the $I_{h}$ group based on group theory. The results are listed in the Appendix A. 

In the calculation of the imaginary part of dielectric response function, $\varepsilon_{2}\left(\omega \right)$, we have introduced a broadening parameter γ. Namely, the delta function in Equation (29) is replaced by a Lorentz function $\frac{\gamma /\pi }{{\left(\hbar \omega +E_{v,i}-E_{c,j}\right)}^{2}+\gamma^{2}}$. $\varepsilon_{2}\left(\omega \right)$ of a C_60_ buckyball calculated by the current method is shown in Figure 2 together with the corresponding results obtained by using the QE package. Our results are in excellent agreement with the QE results on the low-energy side with ħω < 6 eV. For photon energies higher than 6 eV, we still get similar spectral features with roughly the same average oscillator strengths, but the details are somewhat different. The main deviation is due to the difference in boundary conditions used between our approach and the QE package. Here, we consider an isolated C_60_ buckyball confined by an infinite potential at a radius of 12 bohrs (imposed by the cut-off of B-spline functions), while the QE package adopts a supercell with periodic boundary condition. Different boundary conditions will lead to different dielectric response functions at high photon energies [41,42]. Since all the eigenstates of well-defined symmetry have been obtained, the computation of the dipole strength based on Equation (31) can be calculated very efficiently (with less than 10 seconds) when the selection rule and Wigner–Ekart theorem are adopted. If we perform the calculation by a brute-force method without considering the symmetry, it will take much longer. The same concept can be applied to the calculation of the excitonic effect for the C_60_ buckyball by solving the Bethe–Salpeter equation [21]. It is expected that the use of symmetrized basis can also speed up the computation significantly in comparison to the brute-force method.

## 4. Conclusions

Using the symmetrized-basis approach, we have implemented a highly efficient DFT code for the C_60_ buckyball. The energy levels calculated by this method are in close agreement with those obtained by using the Quantum Espresso (QE) package and in fair agreement with results obtained by the all-electron calculation with Gaussian 16 package. The computation time needed to obtain the self-consistent charge density of C_60_ buckyball is about 1/3 of that by using the QE package. Note that our code is not yet fully optimized, and therefore, it has the potential to speed up further. Once the self-consistent charge density is obtained, the computation of 120 occupied levels and 300 unoccupied levels with the current code takes only about 300 s, while it would take more than 100 times longer to do the same by using the QE package. For the calculation of the optical excitation spectrum from the 120 occupied levels to 300 unoccupied levels (not including the excitonic effect), the CPU time needed is less than 10 s after obtaining the eigenstates. 

The method can be readily extended to other fullerenes such as C_70_ and C_80_ within the geometry-adapted symmetrized basis set (with reduced numbers of symmetry operations). Here, we have introduced a simple scheme to utilize the computation method to extract the coefficients in the symmetrized basis that transform according to the IRs of the underlying point group as described in Section 2.3. Thus, applying the same idea to other fullerenes can be conveniently achieved. This method can also be extended to study fullerene-like crystals and fullerene-related quantum devices. For such application, we will calculate the coupling matrix describing the interaction of the fullerene with neighboring objects based on the first principles in the framework of the linear combination of molecular orbitals (LCMO) approach. Since we use symmetry-adapted harmonics augmented by B-splines as basis functions, all eigenfunctions of the systems considered are localized near the fullerene surface, this can be done conveniently, and the resulting Hamiltonian matrix will be sparse. 

For optoelectronic properties, the exciton plays a significant role. A model calculation of excitonic states in C_60_ crystals based on the LCMO method has been reported, in which the MOs are deduced from a DFT-GW calculation [43]. The overlap integrals of MOs between two adjacent C_60_ molecules have been neglected. Here, with the use of icosahedral harmonics augmented by B-splines as basis functions, the intra-molecular optical transition matrix and electron-hole Coulomb scattering matrix can be computed with very little effort. Thus, we can calculate the effect of inter-molecular overlap integrals on the excitonic states efficiently with the current approach. 

## Figures and Tables

**Figure 1 nanomaterials-13-01912-f001:**
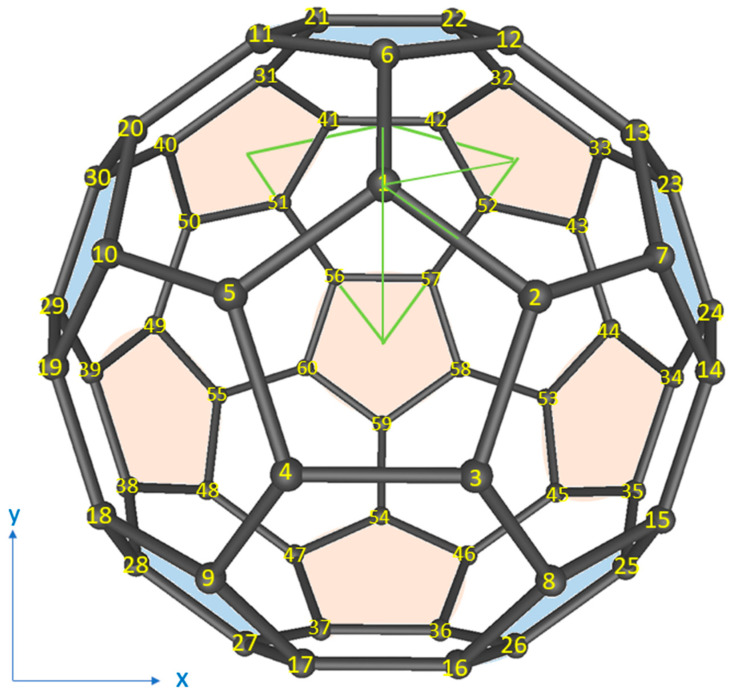
Schematic diagram of buckyball [25].

**Figure 2 nanomaterials-13-01912-f002:**
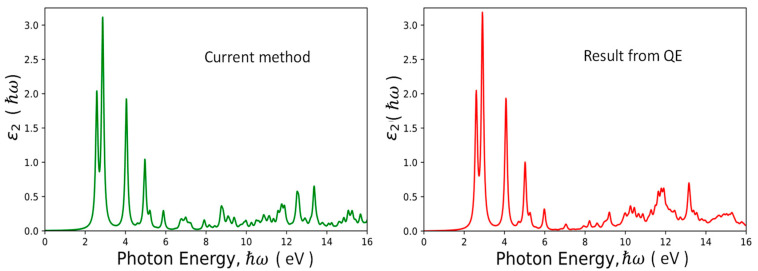
Imaginary part of dielectric response function, $\varepsilon_{2}\left(\hbar \omega \right)$ of the C_60_ buckyball calculated by the current method. The broadening parameter (*Γ*) used is 0.005 eV.

**Table 1 nanomaterials-13-01912-t001:** The symmetrized basis functions with the lowest *l* for the five irreducible representations of the $I_{h}$ group with either even (g) or odd (u) parity.

$\Gamma_{1g}$	$Y_{0,0}$
$\Gamma_{2g}$	$\sqrt{2}Im\left\{\sqrt{33}\;Y_{6,1}-\sqrt{11}\;Y_{6,4}-\sqrt{6}\;Y_{6,6}\right\}/\sqrt{50},$ $\sqrt{2}\;{ReY}_{6,5},$ $\sqrt{2}Re\left\{\sqrt{33}\;Y_{6,1}+\sqrt{11}\;\;Y_{6,4}-\sqrt{6}\;Y_{6,6}\right\}/\sqrt{50}$
$\Gamma_{3g}$	$\sqrt{2}Re\left\{{\sqrt{286}Y}_{8,2}+\sqrt{39}\;Y_{8,3}+\sqrt{63}\;\;Y_{8,7}-\sqrt{112}\;Y_{8,8}\right\}/\sqrt{500}$ ${\sqrt{2}ImY}_{8,5},\sqrt{2}Im\left\{{\sqrt{286}Y}_{8,2}-\sqrt{39}\;Y_{8,3}+\sqrt{63}\;Y_{8,7}+\sqrt{112}\;Y_{8,8}\right\}/\sqrt{500}$
$\Gamma_{4g}$	$\sqrt{2}Re\left\{\sqrt{14}Y_{4,2}+Y_{4,3}\right\}/\sqrt{15},$ $\sqrt{2}Im\left\{\sqrt{14}Y_{4,2}-Y_{4,3}\right\}/\sqrt{15}$ $\sqrt{2}Re{\{\sqrt{7}Y}_{4,1}-\sqrt{8}Y_{4,4}\}$,$\;\sqrt{2}Im{\{\sqrt{7}Y}_{4,1}+\sqrt{8}Y_{4,4}\}/\sqrt{15}$
$\Gamma_{5g}$	$Y_{2,0},\sqrt{2}ReY_{2,1},\;\sqrt{2}\;ImY_{2,1},\;\sqrt{2}ReY_{2,2},\;\sqrt{2}ImY_{2,2}$
$\Gamma_{1u}$	$\sqrt{2}\{\sqrt{3335}{ReY}_{15,5}+\sqrt{1914}{ImY}_{15,9}+\sqrt{1001}{ReY}_{15,14}\}/\sqrt{6250}$
$\Gamma_{2u}$	$Y_{1,0},\;\sqrt{2}ReY_{1,1},\sqrt{2\;}ImY_{1,1}$
$\Gamma_{3u}$	$Y_{3,0},$ $\sqrt{2}Re\left\{\sqrt{3}Y_{3,2}-\sqrt{2}Y_{3,3}\right\}/\sqrt{5},\sqrt{2}\;Im\left\{\sqrt{3}Y_{3,2}+\sqrt{2}Y_{3,3}\right\}/\sqrt{5}$
$\Gamma_{4u}$	$\sqrt{2}ReY_{3,1},\sqrt{2}\;ImY_{3,1},\sqrt{2}\;Re\left\{\sqrt{2}Y_{3,2}+\sqrt{3}Y_{3,3}\right\}/\sqrt{5}$ $,\sqrt{2}\;Im\left\{\sqrt{2}Y_{3,2}-\sqrt{3}Y_{3,3}\right\}/\sqrt{5}$
$\Gamma_{5u}$	$\sqrt{2}Re\left\{{\sqrt{2}\;Y}_{5,2}-\sqrt{3}\;Y_{5,3}\right\}/\sqrt{5},$ $\sqrt{2}Im\left\{{\sqrt{2}\;Y}_{5,2}+\sqrt{3}Y_{5,3}\right\}/\sqrt{5}$ $\sqrt{2}\;{ImY}_{5,5},\sqrt{2}\;Re\left\{\sqrt{7}Y_{5,1}-\sqrt{3}\;Y_{5,4}\right\}/\sqrt{10},\sqrt{2}Im\left\{\sqrt{7}\;Y_{5,1}+\sqrt{3}Y_{5,4}\right\}/\sqrt{10}$

**Table 2 nanomaterials-13-01912-t002:** The number of basis states for each symmetry type.

IR	$\Gamma_{1g}$	$\Gamma_{1u}$	$\Gamma_{2g}$	$\Gamma_{2u}$	$\Gamma_{3g}$	$\Gamma_{3u}$	$\Gamma_{4g}$	$\Gamma_{4u}$	$\Gamma_{5g}$	$\Gamma_{5u}$
B-spline	18	18	18	18	18	18	18	18	18	18
SABF	23	13	46	60	46	60	69	72	92	84
Total	414	234	828	1080	828	1080	1242	1296	1656	1512

**Table 3 nanomaterials-13-01912-t003:** The number of occurrences (*n*) and the number of basis states (n×g ) for each symmetry type for occupied energy levels.

IR	$\Gamma_{1g}$	$\Gamma_{1u}$	$\Gamma_{2g}$	$\Gamma_{2u}$	$\Gamma_{3g}$	$\Gamma_{3u}$	$\Gamma_{4g}$	$\Gamma_{4u}$	$\Gamma_{5g}$	$\Gamma_{5u}$
n	3	0	1	4	1	4	4	4	7	4
n×g	3	0	3	12	3	12	16	16	35	20

**Table 4 nanomaterials-13-01912-t004:** Energy level differences (in eV) of C_60_ molecules. See text for details. $E_{g}$ means the energy difference between the lowest unoccupied molecular orbital (LUMO) (with symmetry $\Gamma_{2u}$) and the highest occupied molecular orbital (HOMO) (with symmetry${\;\Gamma }_{5u}$ ).

Transitions	BAIH	QE	Gaussian
$∆c_{3}$ ** $\;(\Gamma_{5g}-\Gamma_{3u})$ **	0.28	0.30	0.22
$∆c_{2}$ ** $\;(\Gamma_{3u}-\Gamma_{2g})$ **	1.20	1.17	1.32
$∆c_{1}$ ** $\;(\Gamma_{2g}-{\;\Gamma }_{2u})$ **	1.17	1.18	1.13
$E_{g}$**$\;({\;\Gamma }_{2u}-{\;\Gamma }_{5u}$** )	1.43	1.40	1.47
$∆v_{1}$ ** $\;(\Gamma_{5u}-\Gamma_{4g})$ **	1.22	1.23	1.21
$∆v_{2}$ ** $\;(\Gamma_{4g}-\Gamma_{5g})$ **	0.25	0.25	0.25
$∆v_{3}$ ** $\;(\Gamma_{5g}-\Gamma_{5u})$ **	1.47	1.48	1.51
$∆v_{4}$ ** $\;(\Gamma_{5u}-\Gamma_{4u})$ **	0.04	0.04	0.01
$∆v_{5}$ ** $\;(\Gamma_{4u}-\Gamma_{5g})$ **	0.22	0.26	0.30
$∆v_{6}$ ** $\;(\Gamma_{5g}-\Gamma_{3g})$ **	0.63	0.58	0.56
$∆v_{7}$ ** $\;(\Gamma_{3g}-\Gamma_{4u})$ **	0.28	0.29	0.32
$∆v_{8}$ ** $\;(\Gamma_{4u}-\Gamma_{4g})$ **	0.59	0.63	0.66
$∆v_{9}$ ** $\;(\Gamma_{4g}-\Gamma_{3g})$ **	0.15	0.14	0.14
$∆v_{10}$ ** $\;(\Gamma_{3g}-\Gamma_{5g})$ **	0.23	0.18	0.14
$∆v_{11}$ ** $\;(\Gamma_{5g}-\Gamma_{5g})$ **	0.27	0.30	0.33
$∆v_{12}$ ** $\;(\Gamma_{5g}-\Gamma_{4u})$ **	0.65	0.66	0.67
$∆v_{13}$ ** $\;(\Gamma_{4u}-\Gamma_{2u})$ **	0.23	0.19	0.15
$∆v_{14}$ ** $\;(\Gamma_{2u}-\Gamma_{3u})$ **	0.22	0.25	0.28
$∆v_{15}$ ** $\;(\Gamma_{3u}-\Gamma_{1g})$ **	0.39	0.38	0.33
$∆v_{16}$ ** $\;(\Gamma_{1g}-\Gamma_{5u})$ **	0.46	0.48	0.54
$∆v_{17}$ ** $\;(\Gamma_{5u}-\Gamma_{2u})$ **	0.0	0.03	0.06
$∆v_{18}$ ** $\;(\Gamma_{2u}-\Gamma_{4g})$ **	1.01	1.01	0.99
$∆v_{19}$ ** $\;(\Gamma_{4g}-\Gamma_{5g})$ **	1.31	1.31	1.33

**Table 5 nanomaterials-13-01912-t005:** Selection rules for dipole allowed transitions for a C_60_ buckyball.

Occupied	Unoccupied
${\Gamma_{1g}⊗{\;\Gamma }_{2u}}_{\;}$	${\;\Gamma }_{2u}$
${\Gamma_{1u}⊗\Gamma_{2u}}_{\;}$	$\Gamma_{2g}$
$\Gamma_{2g}⊗{\;\Gamma }_{2u}$	$\Gamma_{1u}⊕\Gamma_{2u}⊕\Gamma_{5u}$
$\Gamma_{2u}⊗{\;\Gamma }_{2u}$	$\Gamma_{1g}⊕\Gamma_{2g}⊕\Gamma_{5g}$
$\Gamma_{3g}⊗{\;\Gamma }_{2u}$	$\Gamma_{4u}⊕\Gamma_{5u}$
$\Gamma_{3u}⊗{\;\Gamma }_{2u}$	$\Gamma_{4g}⊕\Gamma_{5g}$
$\Gamma_{4g}⊗{\;\Gamma }_{2u}$	${\;\Gamma }_{3u}⊕\Gamma_{4u}⊕\Gamma_{5u}$
$\Gamma_{4u}⊗{\;\Gamma }_{2u}$	${\;\;\Gamma }_{3g}⊕\Gamma_{4g}⊕\Gamma_{5g}$
$\Gamma_{5g}⊗{\;\Gamma }_{2u}$	${\Gamma_{2u}⊕\Gamma }_{3u}⊕\Gamma_{4u}⊕\Gamma_{5u}$
$\Gamma_{5u}⊗{\;\Gamma }_{2u}$	${\Gamma_{2g}⊕\Gamma }_{3g}⊕\Gamma_{4g}⊕\Gamma_{5g}$

## Data Availability

Theoretical methods and results are available from the authors.

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
