# Peer review of "Density Functional Theory for Buckyballs within Symmetrized Icosahedral Basis"

_nanomaterials, 2023, doi:10.3390/nano13131912_

Round 1

Reviewer 1 Report

While the submitted manuscript presents an original solution to the problem of modelling of C60, this study has also some drawbacks and major revisions are needed. Details are listed below.

Line 24, please delete „The”

At the end of the introduction, the aim of the study should be stated explicitly

Line 63, please delete „The”

Line 66, it should be “density-functional theory”

Line 77, the choice of NCPP must be better justified. I wonder why the Authors haven’t chosen the ultrasoft pseudopotentials, USP,  that were introduced in order to allow calculations to be performed with the lowest possible cutoff energy for the plane-wave basis set, therefore reducing significantly the computational cost of calculations.

Line 184, it should be “local”

Lines 249-254, this part is my major concern. First of all, why the Authors have chosen QE since they are not modeling a periodic structure? For that reason the other codes are the methods of first choice, i.e. Gaussian or DMol. The cost of plane wave calculations is associated with the size of the unit cell, not only with the number and type of atoms inside it. Therefore such comparison, in case of efficiency, is not a proper one. What is more, how can the Authors be sure that the length of 20 Å is enough not to observe the intramolecular forces between the symmetry images? Also, the choice of functional for the QE calculations has not been justified, in fact the name of the functional is not stated either. Besides, was the Ecut value optimized (tested for convergence)?

Line 335, the same figure must be prepared using the results from the other code, in the current version from QE but I strongly recommend to replace it with those from Gaussian, or, add the Gaussian to the QE and compare all three methods.

Lines 384-390, OK, but what about the systems with lower (or no) symmetry? I.e. fullerene derivatives? Or substituted systems? Or clathrates?

English is OK.

Author Response

Reply to comments of Reviewer 1

We thank the reviewer for the critical comments, which help us to improve the manuscript. Our point-to-point replies to the reviewer’s comments are listed below:

Comments: While the submitted manuscript presents an original solution to the problem of modeling of C60, this study has also some drawbacks, and major revisions are needed. Details are listed below.

Line 24, please delete „The”

Reply: The typo has been fixed.

At the end of the introduction, the aim of the study should be stated explicitly

Reply: The following statements have been added at the end of the introduction in the updated version.

“This paper aims to demonstrate the usefulness of symmetrized basis constructed from spherical harmonics and finite-element (B-splines) functions in the development of DFT for high-symmetry systems such as fullerenes and related crystals for application in solid-state devices. The standard analytic procedure for finding suitable linear combinations of spherical harmonic functions based on the group theory can be tedious. In this paper, we introduce a simple scheme to utilize a computation method to extract the coefficients in the symmetrized basis that transform according to the IRs of the underlying point group. It is convenient to use and it can avoid errors introduced by using different conventions for defining the basis functions. We use the C60 buckyball as an example to illustrate the advantages of the method and discuss how to use these convenient basis functions to study related fullerene-like materials.“

Line 63, please delete „The”

Reply: The typo has been fixed.

Line 66, should be “density-functional theory”

Reply: The typo has been fixed.

Line 77, the choice of NCPP must be better justified. I wonder why the Authors haven’t chosen the ultrasoft pseudopotentials, USP,  that were introduced in order to allow calculations to be performed with the lowest possible cutoff energy for the plane-wave basis set, therefore reducing significantly the computational cost of calculations.

Reply: As compared to the ultra-soft pseudopotential (USPP) approach, which uses a projector augmented part to reduce the number of plane waves needed to achieve faster convergence in the calculation, the NCPP has a rather simple form and the coding is much easier. USPP introduces a generalized orthogonal condition (and related terms) to relax the norm-conserving constraint, thus making the coding more cumbersome. Since we aim to demonstrate the usefulness of the symmetrized icosahedral basis based on group theory, it is better to start with a simpler scheme. Future improvement of our code (with more effort) can be made to incorporate the USPP scheme. We expect that with USPP, the number of different Ylm(W) basis functions needed to get convergent results can be reduced significantly. Thus, using USPP should further improve the efficiency of our code. Such discussion on prospect has been included in the revised version. (See Lines 93-99)

Line 184, should be “local”

Reply: The typo has been fixed.

Comment: Lines 249-254, this part is my major concern. First of all, why the Authors have chosen QE since they are not modeling a periodic structure? For that reason, the other codes are the methods of first choice, i.e. Gaussian or DMol.

Reply: The QE package was chosen for comparison because it has the option to adopt the NCPP scheme. Since we also develop the code based on NCPP but with different basis functions, it is natural to compare with results generated from the QE package. Although the C60 molecule considered here is an isolated system, QE is designed for periodic systems. By using a properly chosen supercell, the QE package can also model the properties of an isolated C60 molecule adequately. Furthermore, as eluded in the introduction and conclusions, we intend to apply this method to study the electronic states of crystals made of buckyballs as well. Thus, once we can get good agreement with the results obtained by QE, we are confident that the symmetrized basis for each buckyball adopted here can also be used to construct the crystal basis in the form of linear-combination-of-molecular orbitals (LCMO) approach. When we extend our method to study buckyball crystals, the comparison with the QE result discussed here will be naturally extended to the crystal phase.

Gaussian and DMol. are also good choices for comparison with our approach. However, since the Gaussian package is based on local orbitals centered on atomic sites, the package adopts an all-electron approach, instead of the pseudopotential approach. So, the comparison with our pseudopotential code will not be a fair one. Nonetheless, we have calculated the eigenvalues of the C60 molecule by using the Gaussian 16 package and include the results for inter-level energy spacings in the last column of Table IV in the revised version. The results still show reasonable agreement with ours, but not as good as QE, since the potential used in Gaussian 16 is quite different from QE and ours.

Comment: The cost of plane wave calculations is associated with the size of the unit cell, not only with the number and type of atoms inside it. Therefore, such a comparison, in case of efficiency, is not a proper one. What is more, how can the Authors be sure that the length of 20 Å is enough not to observe the intramolecular forces between the symmetry images? Also, the choice of functional for the QE calculations has not been justified, in fact, the name of the functional is not stated either. Besides, was the Ecut value optimized (tested for convergence)?

Reply: We have checked the suitability of the vacuum length used in the QE calculation and found that the results concerned here are not significantly altered when the cell size is varied between 15 A and 20 A. Since C60 is not electrically polarized, based on previous calculations on graphene nanoribbons [32] that ~10 A of vacuum space is enough. So, a cell length of 16-20 A for C60 is typical (10-14 A  of vacuum + 6 A  for the buckyball). We have clarified this issue in the revised version. Please see the revised version (Lines 276-280).

The exchange-correlation(xc) functional adopted in our QE calculation is based on the parametrized form given in [25] (same as the one used in our code). We also checked the convergence of plane-wave energy cutoff and found that 60-70 Ry is adequate for our purposes. The NCPP option and xc functional adopted in our QE calculation are specified in Lines 271-273 in the revised version

Comment: In line 335, the same figure must be prepared using the results from the other code, in the current version from QE but I strongly recommend to replace it with those from Gaussian, or, add the Gaussian to the QE and compare all three methods.

Reply:

The figure for the imaginary part of the dielectric function obtained by QE is added. (See the right panel of Fig. 2 of the updated manuscript) The agreement between results obtained from our approach and QE are very close.

We also tried to find an option to generate the optical excitation spectrum of the C60 molecule with Gaussian 16 package. However, the only options available include : 1) ZINDO (which is a semiempirical method that was optimized to compute electronic transitions), 2) CIS / CIS(D): Single-excitation configuration interaction is the easiest way to obtain excited states energies., and 3) TD: Time Dependent calculation based on a DFT computational method. TD-DFT obtains the wave functions of MOs that oscillate between the ground state and the first excited states. All three options provide excitation spectra with the inclusion of the electron-hole correlation effect. Thus, they will not provide a suitable comparison with the current simple test (with single-particle approximation). The computation time needed to generate the excitation spectrum is around 15 hours (on a single process) with the ZINDO option (the fastest one among the three options), while with our approach it only takes 5 minutes (including the time to generate 400 eigenstates + optical excitation spectrum). Of course, this is not a fair comparison, since we did not consider the electron-hole correlation effect in the current study. This effect will be considered when we solve the Bethe-Salpeter equation in the future. Thus, it is not adequate to include such a comparison in the current paper.

Comment: Lines 384-390, OK, but what about the systems with lower (or no) symmetry? I.e. fullerene derivatives? Or substituted systems? Or clathrates?

Reply: In this paper, we have introduced a simple scheme to utilize the computation method to extract the coefficients in the symmetrized basis that transform according to the IRs of the underlying point group as described in Sec. 2.3. Thus, applying this symmetrized-basis method to any systems of reduced symmetry can be conveniently done. Even for a system of low symmetry, the idea of using symmetrized basis can still be helpful, since a single symmetry operation can block-diagonalize the Hamiltonian matrix into two decoupled sub-Hamiltonians. Diagonalizing a matrix of half the size via a direct solver will take only 1/8 of the time, not to mention the benefit of saving storage space during the computation. Of course, for systems with no symmetry, the idea of symmetrized basis is not applicable.

Some discussions on how to extend our approach to other fullerenes with lower symmetry are added in the Conclusions section. (See Lines 417-422)

Reviewer 2 Report

This manuscript reports on modeling work on the density functional theory for buckyballs within symmetrized icosahedral basis. The authors approach the problem in a really interesting and original way, while research questions they have investigated and answered are of great theoretical interest and impact. They constructed and tested the corresponding model very well. DFT/basis details and choices with respect to existing approaches are original and innovative, as well as very well documented and presented. Equally important, the model is feasible and practical for the ambitious task of approaching similar systems. The authors also put the emphasis on the good motivation for such a study. Thus, the authors achieved useful and convincing results clearly interesting to a wide audience of readers.

In addition, besides being very timely, the present work is also much credible and highly original.

The presentation of results is clear and appealing, an easy and consistent read and thus can promote for future studies. Useful and employable solutions can be extracted from the present modeling, while the results are given excellent context.

All in all, this work certainly represents a valuable contribution with possible wider impact to the field.

The authors chose an adequate structure of the manuscript. Also, concise, and nicely illustrated figures and their corresponding analysis are provided.

This work once published would be instructive and suggestive in terms of further studies and with excellent chances of being widely cited.

There are some very minor issues with this already excellent manuscript that will need to be addressed before the manuscript becoming suitable for publication, i.e., it can be considered for publication after a minor revision:

1: Title: “buckyballs” should not start with a capital letter. "Densiti functional" is written without hifen.

2: Authors could think to substitute “buckyballs” with “fullerenes”/”fullerene-like material systems”. This is the more modern term and will give a wider outlook to their work.

3: Abstract should mention applicability of the method developed in this work to wider group of systems, something like “fullerenes and fullerene-like materials”.

4: In the introduction, the authors should also mention that their model/method is efficiently applicable even to solid systems which are “fullerene-like” like thin and ultra-thin films, only the appropriate scaling should be applied to structural aspects of systems of similar “fullerene-like” complexity, namely [Chemical Physics Letters 506 (2011) 86-91; Thin Solid Films 515 (2003) 1028-1032]. Such works should be acknowledged thus providing a wider context of the present work.

5: Spell-check and stylistic revision of the paper are necessary. Some long sentences, as well as misspellings, etc., are noticeable throughout the text.

Spell-check and stylistic revision of the paper are necessary. Some long sentences, as well as misspellings, etc., are noticeable throughout the text.

Author Response

Comments: This manuscript reports on modeling work on the density functional theory for buckyballs within the symmetrized icosahedral basis. The authors approach the problem in a really interesting and original way, while research questions they have investigated and answered are of great theoretical interest and impact. They constructed and tested the corresponding model very well. DFT/basis details and choices with respect to existing approaches are original and innovative, as well as very well documented and presented. Equally important, the model is feasible and practical for the ambitious task of approaching similar systems. The authors also put the emphasis on the good motivation for such a study. Thus, the authors achieved useful and convincing results clearly interesting to a wide audience of readers.

In addition, besides being very timely, the present work is also much credible and highly original.

The presentation of results is clear and appealing, an easy and consistent read and thus can promote for future studies. Useful and employable solutions can be extracted from the present modeling, while the results are given excellent context.

All in all, this work certainly represents a valuable contribution with possible wider impact to the field.

Reply: We thank Reviewer 1 for the very positive comments which confirm the originality and feasibility of our approach for investing in buckyballs and fullerene-like materials.

There are some very minor issues with this already excellent manuscript that will need to be addressed before the manuscript becoming suitable for publication, i.e., it can be considered for publication after a minor revision:

1: Title: “buckyballs” should not start with a capital letter. "Density functional" is written without hifen.
Reply: fixed

2: Authors could think to substitute “buckyballs” with “fullerenes”/”fullerene-like material systems”. This is the more modern term and will give a wider outlook to their work.

Reply: Since we use symmetrized icosahedral basis, which is suited for C60 buckyball only, we prefer to keep this title. However, we added a paragraph in the introduction to emphasize that the method introduced here can be easily extended to other fullerene-like materials.

3: Abstract should mention applicability of the method developed in this work to wider group of systems, something like “fullerenes and fullerene-like materials”.

Reply: Revised as suggested. See Line 20 in the abstract of the revised version.

4: In the introduction, the authors should also mention that their model/method is efficiently applicable even to solid systems which are “fullerene-like” like thin and ultra-thin films, only the appropriate scaling should be applied to structural aspects of systems of similar “fullerene-like” complexity, namely [Chemical Physics Letters 506 (2011) 86-91; Thin Solid Films 515 (2003) 1028-1032]. Such works should be acknowledged thus providing a wider context of the present work.

Reply: Revised as suggested. We added a paragraph in Introduction to discuss the aim of the paper and how it can be extended to other fullerene-like materials. The suggested references are added in the revised version as [22,23]. See Lines 63-75.

5: Spell-check and stylistic revision of the paper are necessary. Some long sentences, as well as misspellings, etc., are noticeable throughout the text. Spell-check and stylistic revision of the paper are necessary. Some long sentences, as well as misspellings, etc., are noticeable throughout the text.

Reply: Typos and long sentences are corrected in the revised version. Spellings were checked by using grammarly.

Round 2

Reviewer 1 Report

The Authors have great;ly improved their work since original submission, which I appreciate. While I still can't agree with some ideas and statements, I guess it is ok to have different views on the same topic. Current version can be accepted, I find no major issues to correct.